# Association between Household Exposure to Secondhand Smoke and Dental Caries among Japanese Young Adults: A Cross-Sectional Study

**DOI:** 10.3390/ijerph17228623

**Published:** 2020-11-20

**Authors:** Hikari Saho, Ayano Taniguchi-Tabata, Daisuke Ekuni, Aya Yokoi, Kouta Kataoka, Daiki Fukuhara, Naoki Toyama, Md Monirul Islam, Nanami Sawada, Yukiho Nakashima, Momoko Nakahara, Junya Deguchi, Yoko Uchida-Fukuhara, Toshiki Yoneda, Yoshiaki Iwasaki, Manabu Morita

**Affiliations:** 1Department of Preventive Dentistry, Okayama University Graduate School of Medicine, Dentistry and Pharmaceutical Sciences, Okayama 700-8558, Japan; dekuni7@md.okayama-u.ac.jp (D.E.); de18017@s.okayama-u.ac.jp (K.K.); pu171qxi@s.okayama-u.ac.jp (N.T.); p3a99o50@s.okayama-u.ac.jp (M.M.I.); de422027@s.okayama-u.ac.jp (N.S.); pric37ll@s.okayama-u.ac.jp (M.N.); pmwi1oa1@s.okayama-u.ac.jp (J.D.); de20006@s.okayama-u.ac.jp (Y.U.-F.); de17057@s.okadai.jp (T.Y.); mmorita@md.okayama-u.ac.jp (M.M.); 2Department of Preventive Dentistry, Okayama University Hospital, Okayama 700-8558, Japan; de19026@s.okayama-u.ac.jp (A.T.-T.); yokoi-a1@cc.okayama-u.ac.jp (A.Y.); de20041@s.okayama-u.ac.jp (D.F.); nakashima109@okayama-u.ac.jp (Y.N.); 3Health Service Center, Okayama University, Okayama 700-8530, Japan; yiwasaki@okayama-u.ac.jp

**Keywords:** secondhand smoke, dental caries, permanent dentition, young adult

## Abstract

The long-term effects of secondhand smoke (SHS) on dental caries among Japanese young adults remain unclear. The purpose of this cross-sectional study was to evaluate whether household exposure to SHS is associated with dental caries in permanent dentition among Japanese young adults. The study sample included 1905 first-year university students (age range: 18–19 years) who answered a questionnaire and participated in oral examinations. The degree of household exposure to SHS was categorized into four levels according to the SHS duration: no experience (−), past, current SHS < 10 years, and current SHS ≥ 10 years. Dental caries are expressed as the total number of decayed, missing, and filled teeth (DMFT) score. The relationships between SHS and dental caries were determined by logistic regression analysis. DMFT scores (median (25th percentile, 75th percentile)) were significantly higher in the current SHS ≥ 10 years (median: 1.0 (0.0, 3.0)) than in the SHS—(median: 0.0 (0.0, 2.0)); *p* = 0.001). DMFT ≥ 1 was significantly associated with SHS ≥ 10 years (adjusted odds ratio: 1.50, 95% confidence interval: 1.20–1.87, *p* < 0.001). Long-term exposure to SHS (≥10 years) was associated with dental caries in permanent dentition among Japanese young adults.

## 1. Introduction

Secondhand smoke (SHS), known as environmental tobacco smoke, is a combination of mainstream smoke exhaled by a smoker and sidestream smoke released by the lighted end of a cigarette. As SHS does not pass through a filter, it contains higher concentrations of harmful components than smoke that is inhaled through a cigarette [1,2]. Thus, SHS is related to diseases including coronary heart disease [3,4], stroke [5], lung cancer [6,7], and dental caries [8].

Dental caries is one of the most prevalent infectious diseases. It is defined as the localized destruction of tooth tissue by bacterial action [9]. A recent systematic review [8] suggested an association between SHS and dental caries in both primary and permanent dentitions of children and adolescents. As a mechanism, nicotine promotes the growth of the cariogenic bacterium *Streptococcus mutans* [10]. In addition, SHS causes respiratory tract inflammation [11] and indirectly causes mouth breathing/dry mouth. SHS causes histological changes in salivary glands and salivary composition [12], so caries are more likely to occur by the change in saliva flow rate and function (e.g., self-cleaning action and buffering action against acid). However, compared to available evidence about the effects of SHS on deciduous teeth of children [13], little is known about the long-term effects (≥10 years) of SHS on the permanent dentition of Japanese adolescents and/or young adults (aged > 15 years).

In Japan, tobacco smoke is restricted by the Health Promotion Act. The revised law [14] has encouraged authorities of public places such as hospitals, schools, municipal offices, and restaurants to take the lead in implementing smoking restrictions and in preventing SHS for non-smokers, especially in populations younger than 20 years. However, household exposure to SHS is not regulated. Therefore, it is important to explore the effects of household SHS and to prevent household exposure to SHS.

Here, we focused on the effects of household exposure to SHS among Japanese adolescents and young adults. In this study, we hypothesized that household exposure to SHS is a risk factor for dental caries in permanent dentition among Japanese young adults. Therefore, the aim of this cross-sectional study was to evaluate whether household exposure to SHS is associated with dental caries in permanent dentition among Japanese young adults.

## 2. Materials and Methods

### 2.1. Study Population

In this cross-sectional study, data were collected from first-year university students who answered the questionnaire and underwent oral examinations at the Health Service Center of Okayama University in April 2019. The inclusion criteria were young adults between 18 and 19 years of age. The exclusion criteria for analysis were: (i) aged ≥20 years, (ii) being a current smoker, and/or (iii) providing incomplete data. We estimated the sample size using G*Power version 3.1 statistical software (Universität Kiel, Kiel, Germany). We hypothesized an odds ratio of 1.3 by logistic regression analysis by referring to a past study [8] with a power of 80% and a two-sided significance level of 5%. Thus, we needed to include a minimum of 386 participants.

### 2.2. Ethical Procedures and Informed Consent

This study protocol was approved by the Ethics Committees of Okayama University Graduate School of Medicine, Dentistry and Pharmaceutical Sciences and Okayama University Hospital (no. 1060). Verbal and written informed consent to participate in the study were provided by all selected participants. The study’s reporting conforms to STROBE guidelines (Appendix A).

### 2.3. Questionnaire

The questionnaires addressed age, sex, daily frequency of tooth brushing, use of dental floss, regular dental checkups, consumption frequency of sugar-sweetened snacks and/or soft drinks, experience of topical fluoride application, knowledge about dietary education, and household exposure to SHS. The following factors were evaluated by yes or no answers: use of dental floss [15], regular dental checkups [15], experience of topical fluoride application [16], and knowledge of dietary education [17]. Daily frequency of tooth brushing (≤1 time, 2 times, or ≥3 times) [15] and consumption frequency of sugar-sweetened snacks and/or soft drinks (never, 1 or 2 times per day, or ≥3 times per day) [18] were evaluated in the form of ternary answers. Questions on household exposure to SHS included two parts: (i) Does anyone in your household smoke? (current smokers, past smokers, or no smokers) and (ii) If there are any smokers in your household, have you been exposed to SHS for greater than or equal to 10 years? (yes or no) based on the previous decision to a 10-year threshold for the exposure categories [19]. Household exposure to SHS was then categorized into four levels: (i) no experience of SHS, SHS −; (ii) past experience of household exposure, past SHS; (iii) current household exposure to SHS < 10 years, current SHS < 10 years; and (iv) current household exposure to SHS ≥ 10 years, current SHS ≥ 10 years.

### 2.4. Oral Examination

Nine qualified dentists (D.E., A.T.-T., H.S., A.Y., T.Y., N.T., Y.U.-F., D.F., and K.K.) conducted oral examinations. The World Health Organization diagnostic criteria for dental caries experience [20] were used to evaluate decayed, missing, and filled teeth (DMFT). Oral hygiene status was evaluated by visually inspecting selected teeth based on the Debris Index-Simplified (DI-S) [21]. A modified version of the Index of Orthodontic Treatment Need [22] was used to evaluate malocclusion. The presence of an orthodontic appliance in the oral cavity was visually checked. Qualified dentists repeatedly practiced DMFT scoring and malocclusion evaluation in five volunteers for two weeks. For the oral examination, intra- and inter-agreement were good (Kappa statistic > 0.8).

### 2.5. Statistical Analyses

The normality of data was investigated by the Shapiro–Wilk test. We did not confirm the normal distribution of each value. The Kruskal–Wallis test and Dunn test with Bonferroni correction were used to compare DMFT scores by SHS status. The chi-squared test or Mann–Whitney *U* test was used to examine the difference between groups with (DMFT ≥ 1) and groups without (DMFT = 0) dental caries. The association between dental caries (DMFT ≥ 1) and independent variables that are risk factors of dental caries was analyzed using logistic regression. Independent variables were selected when the *p*-value was <0.20 for the chi-square test or unpaired *t*-test in each variable and were based on previous studies because it has been suggested that potential confounders should be eliminated only if *p* > 0.20 to prevent residual confounding [23]. All analyses were performed using SPSS^®^ 21.0 J software (IBM Japan Ltd., Tokyo, Japan). Significant differences were considered to exist at *p* < 0.05.

## 3. Results

A total of 2241 participants answered the questionnaire and underwent oral examinations in 2019. Of these students, 336 students were excluded for the following reasons: (i) being aged ≥ 20 years (*n* = 114), (ii) being a current smoker (*n* = 4), and (iii) providing incomplete data (*n* = 218). Finally, 1905 participants (85.0%) were analyzed.

The characteristics of the participants are shown in Table 1. Overall, 1101 (57.8%) participants were men. The median (25th percentile, 75th percentile) DMFT score was 0.0 (0.0, 2.0). For SHS status, 438 (23.0%) participants answered that they experienced long-term household exposure to SHS (≥10 years).

Figure 1 compares DMFT scores by household exposure to SHS (box-plot). DMFT scores in the SHS ≥10 years category (median: 1.0 (0.0, 3.0)) were significantly higher than those in the SHS − group (median; 0.0 (0.0, 2.0)) (*p* = 0.001).

Table 2 shows the comparison between the DMFT = 0 and DMFT ≥ 1 groups. The DMFT ≥ 1 group included 884 (46.4%) participants. Significant differences in daily frequency of tooth brushing and household exposure to SHS were observed between the two groups (*p* < 0.05 for all).

In Table 3, logistic regression analysis revealed that the risk of DMFT ≥1 was significantly associated with use of an orthodontic appliance (odds ratio (OR), 1.45; 95% confidence interval (CI), 1.02–2.07, *p* = 0.039), daily frequency of tooth brushing (≤1 time) (OR, 1.76; 95% CI, 1.22–2.54, *p* = 0.002), and household exposure to SHS ≥ 10 years (OR, 1.50; 95% CI, 1.20–1.87, *p* < 0.001). Significant associations between the other variables and DMFT ≥ 1 were not observed.

## 4. Discussion

In this study, we hypothesized that household exposure to SHS is a risk factor of dental caries in permanent dentition among Japanese young adults. To the best of our knowledge, this is the first study to show the long-term effects (≥10 years) of household exposure to SHS on dental caries of permanent dentition among Japanese young adults between the ages of 18 and 19. This study found that household exposure to SHS ≥10 years is associated with dental caries (OR, 1.50; 95% CI, 1.20–1.87, *p* < 0.001). The findings of this study support those of a previous meta-analysis [8] that indicated an association between caries in permanent dentition and postnatal exposure to SHS in children and adolescents (OR, 1.30; 95% CI, 1.25–1.34, *p* < 0.001).

Our results suggest that household exposure to SHS should be considered in the prevention of dental caries in young adults. Avoiding exposure to SHS is meaningful not only for the prevention of systemic diseases but also for dental caries. Therefore, an early-stage approach should be used to minimize household exposure to SHS and to prevent dental caries before children enter university.

Increasing evidence supports a causal role for SHS in caries formation [24]. A cross-sectional study showed that children exposed to SHS had a higher prevalence of dental caries, higher salivary levels of cariogenic bacteria (e.g., *Streptococcus mutans* and *Lactobacillus*), lower salivary pH, lower flow rate, and lower buffering capacity than non-exposed children [24]. Furthermore, a review suggested that saliva from household members, such as parents or grandparents, affects the bacteria responsible for dental caries’ formation in their children/grandchildren [25]. An in vitro study showed that nicotine promoted *S. mutans* growth [12]. Therefore, children who live with a family member who smokes may be more likely to be transmitted these bacteria [12]. As SHS is known to increase respiratory tract inflammation and cause respiratory illnesses including asthma and allergic rhinitis [11], it can indirectly cause mouth breathing and thus result in dry mouth by an effective decrease in saliva. In addition, an animal study found that histological changes in salivary glands were observed, and decreases in total protein amount, amylase activity, and peroxidase activity in saliva were detected among SHS-exposed rats compared to non-exposed rats [12]. Saliva, acting as a buffering agent against acid, physically removes debris from tooth surfaces and has immunological and bacteriostatic properties [26]. Therefore, SHS may reduce the protective properties of saliva against dental caries. Other cross-sectional studies suggested an association between SHS and vitamin C in dental caries of children [27,28]. SHS is associated with decreased serum vitamin C levels [27], which, in turn, are associated with the growth of cariogenic bacteria [28]. Taken together, SHS could promote dental caries through effects on cariogenic bacteria and salivary condition.

In this study, the major component of DMFT score was filled teeth. Since DMFT means the past experiences of caries, the present effect of SHS on caries remains unclear. Only 156 students (8.2%) had one or more decayed teeth in the present study population. We analyzed the impact of SHS on decayed teeth to investigate the present effect of SHS on dental caries. As a result, we found no significant relationship between SHS and decayed teeth (data not shown). Thus, the present effect of SHS on present decayed teeth in this age group may be minor.

Our results showed that use of an orthodontic appliance is one of the risk factors for dental caries. This finding supports those of a recent review that suggested an association between use of an orthodontic appliance and the risk of dental caries [29]. Thus, clinicians should educate young adults about the susceptibility to dental caries and should educate young adults with orthodontic appliances about careful tooth cleaning practices.

Our results also showed that daily frequency of tooth brushing one or fewer times is a risk factor for dental caries. A cohort study showed that higher brushing frequency is associated with lower decayed and filled surface increases among participants aged 9, 13, and 17 years in the United States [30]. These findings suggest that more frequent fluoride exposure from dentifrice and mechanical disruption of cariogenic biofilms contribute to caries prevention.

We did not identify significant associations between dental caries experience and other risk factors. No significant difference in sex was found between DMFT = 0 and DMFT ≥ 1 groups. Some reviews [31,32,33] suggested effects of sex difference on dental caries. However, there was heterogeneity and further investigation is needed. No significant difference in DI-S was found between DMFT = 0 and DMFT ≥ 1 groups in this study. As we could not investigate oral hygiene status in the full mouth and fundamental differences in dental plaque ecology as a risk of dental caries [34], careful attention is needed regarding the interpretation of these results. Similar to results of our previous study [17], frequent consumption of sugar-sweetened snacks and/or soft drinks was not associated with dental caries. The effects of sugar consumption on dental caries may be low among young adults. We found no association between the experience of topical fluoride application and dental caries. One potential reason for this lack of association may be because we only investigated the experience of topical fluoride application, not the frequency and period. Moreover, the cutoff value (DMFT = 0 and DMFT ≥ 1) may have affected these results. The lack of associations with these established dental caries risk factors in the present study should be considered a limitation when evaluating all other reported study findings, including those for SHS.

There are some limitations associated with the present study. First, all participants were enrolled from a single institution. The percentages of participants who brush their teeth more than one time per day and who experienced topical fluoride application and a DMFT score reported in this study were superior to data reported from a national survey in Japan (89.7% vs. 77.0%, 65.5% vs. 62.5%, and 1.6 vs. 3.1, respectively) [35]. Therefore, extrapolating these results to the general Japanese population might be limited. Second, salivary/serum cotinine level [36], salivary flow rate [24], salivary buffering capacity [24], salivary pH [24], and salivary levels of cariogenic bacteria (e.g., *Streptococcus mutans* and *Lactobacillus*) [24] were not examined in this study. We did not investigate other qualitative assessments (e.g., number of cigarettes smoked per day, smoking location, and number of smokers) for SHS because we focused on the period of exposure to SHS. These factors might have affected our results. Third, we did not consider possible confounders such as different household situations before entrance, social capital [37] or socioeconomic status [38]. Future studies are needed to assess the effects of these factors. Fourth, we did not include daily use of toothpaste with fluoride in our questionnaires, although fluoride is one of the most important factors for the prevention of caries. In 2015, the market share of toothpaste with fluoride in Japan was 91% [39]. Thus, we assumed that most students in this study use toothpaste with fluoride. Fifth, we did not include the frequency of food in general in our questionnaires, although we focused on consumption frequency of sugar-sweetened snacks and/or soft drinks. Finally, a causal relationship could not be shown because this was a cross-sectional study. Therefore, a cohort study is needed to identify SHS as a direct risk factor for the development of dental caries.

## 5. Conclusions

In conclusion, we found that long-term exposure to household SHS (≥10 years) was associated with dental caries in permanent dentition of Japanese young adults between 18 and 19 years of age. Our results suggest that household exposure to SHS should be taken into consideration in prevention of dental caries in young adults, although further investigation is needed to confirm the association.

## Figures and Tables

**Figure 1 ijerph-17-08623-f001:**
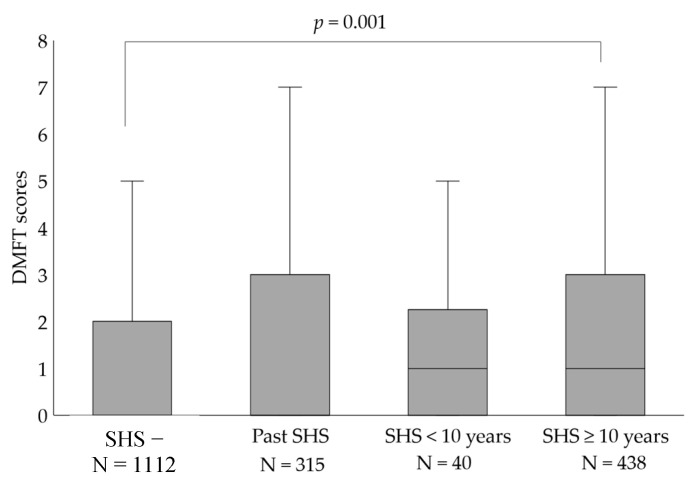
Comparison of DMFT by the presence/absence of household exposure to secondhand smoke. DMFT scores were significantly higher in the current SHS ≥ 10 years than those in the SHS −. Kruskal–Wallis test and Dunn test with Bonferroni correction. DMFT, decayed, missing and filled teeth; SHS, secondhand smoke.

**Table 1 ijerph-17-08623-t001:** Characteristics of participants (*n* = 1905).

Variable		Median (25th percentile, 75th percentile) or *n* (%)
Age (years)		18.0 (18.0, 18.0)
Sex	Male	1101 (57.8)
DMFT		0.0 (0.0, 2.0)
Decayed teeth		0.0 (0.0, 0.0)
Missing teeth		0.0 (0.0, 0.0)
Filled teeth		0.0 (0.0, 2.0)
DI-S		0.2 (0.0, 0.7)
Malocclusion	Yes	673 (35.3)
Use of orthodontic appliance	Yes	135 (7.1)
Daily frequency of tooth brushing	≤1 time	197 (10.3)
	2 times	1395 (73.2)
	≥3 times	313 (16.4)
Use of dental floss	Yes	554 (29.1)
Regular dental checkups	Yes	632 (33.2)
Consumption frequency of sugar-sweetened snacks and/or soft drinks	Never	637 (33.4)
	1 time per day	930 (48.8)
	2 times per day	246 (12.9)
	≥3 times per day	92 (4.8)
Experience of topical fluoride application	Yes	1253 (65.8)
Knowledge about dietary education	Yes	695 (36.5)
Household exposure to SHS	− (no experience)	1112 (58.4)
	Past	315 (16.5)
	Current <10 years	40 (2.1)
	Current ≥10 years	438 (23.0)

DMFT, decayed, missing and filled teeth; DI-S, Debris Index-Simplified; SHS, secondhand smoke.

**Table 2 ijerph-17-08623-t002:** Differences in variables between DMFT = 0 and DMFT ≥1 groups.

Variables ^a^		DMFT = 0 *n* = 1021	DMFT ≥ 1 *n* = 884	*p*-Value ^b, c^
Sex	Male	602 (59.0)	499 (56.4)	0.268
	Female	419 (41.0)	385 (43.6)	
DI-S		0.2 (0.0, 0.7)	0.3 (0.0, 0.7)	0.134
Malocclusion	Yes	360 (35.3)	313 (35.4)	0.946
Use of orthodontic appliance	Yes	62 (6.1)	73 (8.3)	0.064
Daily frequency of tooth brushing	≤1 time	89 (8.7)	108 (12.2)	0.014
	2 times	749 (73.4)	646 (73.1)	
	≥3 times	183 (17.9)	130 (14.7)	
Use of dental floss	Yes	292 (28.6)	262 (29.6)	0.619
Regular dental checkups	Yes	330 (32.3)	302 (34.2)	0.395
Consumption frequency of sugar-sweetened snacks and/or soft drinks	Never	350 (34.3)	287 (32.5)	0.420
	1 or 2 times per day	627 (61.4)	549 (62.1)	
	≥3 times per day	44 (4.3)	48 (5.4)	
Experience of topical fluoride application	Yes	670 (65.6)	583 (66.0)	0.880
Knowledge about dietary education	Yes	372 (36.4)	323 (36.5)	0.963
Household exposure to SHS	− (no experience)	636 (62.3)	476 (53.8)	0.002
	Past	159 (15.6)	156 (17.6)	
	Current <10 years	18 (1.8)	22 (2.5)	
	Current ≥10 years	208 (20.4)	230 (26.0)	

^a^ Data are expressed as median (25th percentile, 75th percentile) or *n* (%). ^b^ Mann–Whitney *U* test, as appropriate. ^c^ Chi-squared test, as appropriate. DMFT, decayed, missing and filled teeth; DI-S, Debris Index-Simplified; SHS, secondhand smoke.

**Table 3 ijerph-17-08623-t003:** Logistic regression analysis when independent variables were selected based on the *p*-value (<0.20) by the chi-square test or Mann–Whitney *U* test.

Independent Variables		ORs	95% CIs	*p*-Value
DI-S		1.11	0.87–1.42	0.402
Use of orthodontic appliance	No	Ref.		
	Yes	1.45	1.02–2.07	0.039
Daily frequency of tooth brushing	≥3 times	Ref.		
	2 times	1.24	0.97–1.60	0.089
	≤1 time	1.76	1.22–2.54	0.002
Household exposure to SHS	− (no experience)	Ref.		
	Past	1.31	1.01–1.68	0.039
	Current <10 years	1.72	0.91–3.25	0.095
	Current ≥10 years	1.50	1.20–1.87	<0.001

Ref., reference group; DI-S, Debris Index-Simplified; SHS, secondhand smoke; OR, odds ratio; CI, confidence interval.

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
