# Peer review of "Association between Household Exposure to Secondhand Smoke and Dental Caries among Japanese Young Adults: A Cross-Sectional Study"

_ijerph, 2020, doi:10.3390/ijerph17228623_

Round 1

Reviewer 1 Report

This study is surely interesting, however some point needs to revise before acceptance.

  1. The authors should mention why non-parametric analyses were used. Additionally, if Fig.1 was indicated by box-plot, the authors should describe it.
  2. All parameters were simultaneously used as independent variables at logistic analysis. The authors tried univariable analysis? It would be better that the independent variables were selected, not everything. The statistical analyses would be better to reconsider.
  3. “Daily use of toothpaste with fluoride” is one of the most important factor for prevention of caries. The authors did not ask as questionnaires?

Reviewer 2 Report

This paper investigates the effects of household exposure to SHS among Japanese university students concerning dental caries. They concluded that long-term exposure (≥ 10 years) to SHS has an impact on dental caries of permanent dentition among Japanese university students.

Introduction:

The authors need to better explore the hypotheses that support the biological plausibility of the association between SHS and caries and explain reasonable physiological associations between these factors in the pathogenesis of the disease.

Methods:

The subjects were students from the university. This is a biased sample because we would expect that those individuals have better educational and socio-economical levels, and this could explain the low DMFT scores. Does not reflect the oral health of the general population (which shows a mean DMFT index of 3.1 among 15–24 years individuals - https://www.mhlw.go.jp/toukei/list/62-17.html).

This research included a large sample of subjects, who fulfilled a questionnaire and underwent dental examinations. The researchers should have included other easily collectible biological information, such as salivary pH, buffer capacity, saliva flow and S mutans and Lactobacilli levels, to verify if long term SHS affects the biological factors associated with the disease in the studied population.

Discussion:

The subjects showed the past experience of caries (since the only component in the DMFT score was “filled”). This needs to be discussed, showing no present effect between SHS and the disease (the ‘D” component of the index).

Reviewer 3 Report

The present study is undoubtedly actual and deserves to be published since it touches on the poorly studied topic of the negative impact of passive tobacco smoking on the intensity of caries.

However, when discussing the hypothesis, the authors refer mainly to one study showing the effect of nicotine on the growth of one of the cariogenic microbes [Huang R, Li M, Gregory RL. Effect of nicotine on growth and metabolism of Streptococcus mutans. Eur J Oral Sci. 2012, 120 (4), 319–325. doi: 10.1111 / j.1600-0722.2012.00971.x.]

The rest of the risk factors mentioned by the authors (the effect of nicotine and smoking on the decrease in the quantity and quality of salivary fluid), were described as pathogenetic factors for direct smoking, and not for the passive one. Moreover, the pathogenetic impact of smoking smokers, according to the literature, contributes to the development of periodontal disease, but not caries.

I would like to know the opinion of the authors of the article on this issue and, if it is possible, to include the discussion of this aspect in the material of the manuscript.

Moreover, in the questionnaire used, the authors only asked questions about the frequency of brushing their teeth during the day but did not address the type of toothpaste that the study participants used. With that, it is well known that fluoride-containing toothpaste (with 1450 ppm fluoride and above) helps prevent the development of caries. It is also considered that the development of caries is influenced not only by the intake of sugary foods/drinks but also by the frequency of food in general. This question was not included in the questionnaire. Why these two most important points were not considered when compiling the questionnaire?

Round 2

Reviewer 2 Report

The manuscript was improved, specially the last paragraph from the discussion, addressing the limitations of the present study.